# Knowledge gaps and structural barriers to prescribing pre-exposure prophylaxis among healthcare providers in North Louisiana: A cross-sectional study

Deborah G. Smith[1]*, Camila Castro[2], Patricia Pichilingue-Reto[3], Catherine G. Dean[4], Alexandre Malek[5], Luis Enrique Espinoza[6]

1 School of Health Professions and Sciences, Louisiana State University Health Sciences Center-Shreveport, Shreveport, Louisiana, United States of America, 2 Departamento de Odontologia, Centro Universitário Christus (UNICHRISTUS), Fortaleza, CE, Brazil, 3 Department of Medicine, Louisiana State University Health Sciences Center-Shreveport, Shreveport, Louisiana, United States of America, 4 School of Medicine, Louisiana State University Health Sciences Center-Shreveport, Shreveport, Louisiana, United States of America, 5 Department of Medicine, Division of Infectious Diseases, Louisiana State University Health Sciences Center-Shreveport, Shreveport, Louisiana, United States of America, 6 College of Nursing and Health Sciences, Texas A&M University – Corpus Christi, Corpus Christi, Texas, United States of America

* deborah.smith@lsuhs.edu

## Abstract

### Background

Pre-exposure prophylaxis (PrEP) uptake remains low in the Southern United States, despite disproportionate HIV burden. This study aimed to assess knowledge, attitudes, and structural barriers influencing PrEP prescribing among healthcare providers in North Louisiana.

### Methods

We conducted a cross-sectional survey from January 2024 to April 2025 among 102 healthcare providers at a medical school and an affiliated hospital in North Louisiana. The survey assessed knowledge of PrEP, prescribing practices, and the perceived barriers. Descriptive statistics and multivariable logistic regression analysis were performed.

### Results

Among 102 healthcare providers, only 27.5% had ever prescribed PrEP, and 48% had never discussed it with patients. The most frequently reported barriers to prescribing PrEP were lack of provider training (84.3%) and absence of clinical guidelines (68.6%). Multivariable analyses showed that providers with higher self-rated PrEP knowledge were significantly more likely to discuss PrEP with patients (adjusted odds ratio [aOR]: 5.10; 95% confidence interval [CI]: 2.31–11.27) and to prescribe PrEP (aOR: 3.29; 95% CI: 1.72–6.30).

**Data availability statement:** All relevant data are within the manuscript and its Supporting Information files.

**Funding:** The author(s) received no specific funding for this work.

**Competing interests:** The authors have declared that no competing interests exist.

## Conclusion

Significant gaps in provider knowledge and institutional readiness persist in high-burden Southern settings. Targeted provider training and supportive institutional policies are essential to improving PrEP implementation.

## Introduction

HIV remains a major global public health challenge, with nearly 40 million people living with HIV worldwide in 2023 and approximately 1.3 million new infections occurring annually. In the United States, nearly 38,000 new HIV diagnoses are reported each year, with the highest burden concentrated in the South [1]. In Louisiana alone, over 23,000 people are living with HIV, with nearly half progressing to an AIDS diagnosis [2]. Pre-exposure prophylaxis (PrEP) has proven to be a safe and effective intervention to prevent HIV infections. However, its adoption remains suboptimal, particularly in regions with elevated HIV prevalence. Although the national PrEP-to-Need Ratio (PnR) has risen from 10 in 2021–14 in 2023, regional disparities persist across the United States. The South had the lowest regional PnR in the U.S. despite making up 53% of new HIV diagnoses in 2022 and only 39% of people using PrEP [3]. These disparities are driven by socioeconomic and structural factors, such as poverty, limited access to healthcare, and scarce resources [4].

While there is a wealth of evidence on the effectiveness of PrEP in preventing HIV transmission, healthcare providers' involvement in PrEP prescribing remains insufficient. Several studies have identified substantial gaps in provider knowledge, attitudes, and prescribing practices as barriers to the widespread PrEP implementation. A cross-sectional survey conducted among primary care providers across ten Southern states found key barriers include insufficient training, perceived stigma, inadequate health insurance coverage, and time constraints. Conversely, facilitators included access to educational resources, streamlined insurance processes, and heightened patient motivation [5]. These provider-level barriers have been documented across diverse healthcare systems globally, suggesting that challenges in PrEP implementation extend beyond local contexts [6].

Despite growing literature on PrEP implementation, few studies have examined provider-level knowledge and structural barriers influencing prescribing within high HIV-burden Southern healthcare settings such as North Louisiana. Therefore, this study aims to assess the knowledge, attitudes, and structural barriers influencing PrEP prescribing among healthcare providers in North Louisiana.

## Materials and methods

### Study design and settings

This cross-sectional survey was conducted between January 2024 and April 2025 among healthcare providers at a medical school and its affiliated teaching hospital in North Louisiana. As part of the Southern United States, Louisiana consistently ranks

among the states with the highest HIV diagnosis rates, underscoring the importance of examining PrEP implementation within this high-burden setting [4,5].

## Participants and recruitment

Eligible participants included physician faculty members, residents, and fellows actively involved in patient care from the following departments: internal medicine, emergency medicine, pediatrics, family medicine, and obstetrics and gynecology. Participants were recruited using convenience sampling through institutional listserv email invitations, printed flyers posted in clinical and academic areas, and brief presentations during departmental grand rounds and morning reports. Although convenience sampling was employed within a single academic medical center, an estimated sample size was considered to ensure adequate precision of proportion estimates. Since there were no prior estimates of the proportion of providers reporting the outcomes of interest (e.g., PrEP prescribing), a conservative assumed proportion of 50% was used, as recommended by Lwanga and Lemeshow [7], in standard sample size formulas for estimating proportions in cross-sectional studies, because this produces the largest required sample size. Assuming a 95% confidence level and a margin of error of ±10 percentage points, the minimum estimated sample size required was approximately 96 participants. A total of 104 healthcare providers completed the survey, exceeding the recommended threshold. Incomplete responses were excluded using listwise deletion, resulting in a final analytic sample of 102. Recruitment materials included a Quick Response (QR) code and a direct hyperlink to the electronic survey, allowing providers to access the questionnaire via computer, tablet, or mobile device. Upon accessing the survey, participants were first presented with an informational letter describing the purpose of the study, the voluntary nature of participation, and the anonymous handling of responses. Participants indicated their willingness to participate by selecting either "agree" or "disagree." Those who declined were excluded from the survey. Individuals who agreed completed eligibility screening questions; only eligible participants were permitted to proceed to the full survey, while ineligible respondents were automatically screened out, and the survey was closed.

## Measures

The survey instrument was developed using previously validated items assessing provider knowledge, attitudes, and PrEP prescribing behaviors [8]. Items were adapted to reflect the regional clinical context and study objectives. The questionnaire was pilot-tested among physicians involved in the study to evaluate clarity, readability, and survey flow. Minor revisions were made prior to distribution to improve question wording and overall usability. Physicians who participated in the pilot test were not included in the final analytic sample.

Provider knowledge was assessed using self-reported familiarity with PrEP, awareness of prescribing guidelines, and prior training exposure. Prescribing practices were measured using items assessing whether providers had discussed or prescribed PrEP and their perceived confidence in identifying eligible patients.

The first set of outcomes was based on physician prescribing practices for PrEP. Participants were asked the following five questions: 1) What percentage of your patients do you perceive as significantly at risk for HIV? 2) Have any patients asked about PrEP? 3) Have you discussed PrEP with a patient? 4) Have you ever prescribed PrEP to a patient? and 5) Have you referred a patient to another provider for PrEP? The first question had four possible responses: less than 5%, 6% to 10%, 11% to 15%, and greater than 16%. The remaining questions had yes and no responses.

The second set of outcomes was based on barriers to PrEP prescription. Participants were asked whether they had encountered barriers based on five possible factors: 1) A lack of provider training/education regarding PrEP; 2) A lack of clinic guidelines/protocol for prescribing/monitoring PrEP; 3) A lack of clinical and lab monitoring requirements; 4) Staffing/time constraints related to risk reduction and PrEP adherence counseling (medication knowledge/counseling, adverse effects, etc.); and 5) A lack of insurance coverage and out-of-pocket patient costs for PrEP and related care (e.g., lab work).

To investigate physicians' perceptions of PrEP, participants were asked the following four questions: 1) How effective do you think PrEP is in preventing acquisition of HIV/AIDS?; 2) How would you rate your knowledge of PrEP?; 3) PrEP will promote risky sexual behavior?; and 4) How likely do you think a patient would be to decrease their sexual risk as a result of PrEP use? Responses to the first question, which measured PrEP effectiveness, were "very effective," "moderate," and "slightly effective." The responses were reverse-coded to indicate increasing effectiveness. Responses to the second question, which measured PrEP knowledge, were "poor," "fair," "good," "very good," and "excellent." Responses to the third question, which measured whether PrEP would promote risky sexual behavior, were "agree," "neither agree nor disagree," and "disagree." Responses were reverse-coded to indicate an increasing perception of risky sexual behavior. The last question measured the decrease in sexual risk as a result of PrEP use; its responses were "very likely," "likely," "slightly likely," and "not likely." The responses were reverse-coded to reflect increasing agreement that PrEP use decreases sexual risk. Demographic covariates collected included age, sex, race/ethnicity, specialty, hospital role (faculty, resident, fellow, other), and years in practice.

### Data management and analysis

Survey data were collected electronically using REDCap (Research Electronic Data Capture), a secure web-based platform, and exported for analysis in IBM® SPSS 23.0. Descriptive statistics (frequencies, percentages, means, and standard deviations) were calculated for all variables. Associations between provider characteristics, knowledge, and prescribing practices were examined using multivariable logistic regression models. Adjusted odds ratios (aORs) with 95% confidence intervals (CIs) were reported. Covariates included in regression models were age, sex, race/ethnicity, specialty, role, and years in practice. Linear regression was used where outcomes were continuous.

### Ethical approval

The study protocol was reviewed and approved by the Louisiana State University Health Sciences Center–Shreveport Institutional Review Board (IRB #00002527). The requirement for written informed consent was waived due to the minimal-risk nature of this voluntary, anonymous survey. No personally identifiable information was collected, and all responses were stored on secure, password-protected servers accessible only to the research team.

## Results

### Demographic and professional characteristics

Demographic and professional characteristics of study participants are presented in Table 1. The study included 102 health care providers practicing in North Louisiana across various clinical specialties and training backgrounds.

### PrEP-related practices and reported barriers

As shown in Table 2, 72.5% (n = 74) of participants had never prescribed PrEP, and 48% (n = 49) had never discussed it. Reported barriers were predominantly training-related (Fig 1).

Percentage of healthcare providers who reported discussing PrEP with patients and ever prescribing PrEP.

### Provider knowledge and PrEP prescribing behaviors

Provider perceptions and knowledge of PrEP were strongly associated with prescribing behaviors. Providers with greater knowledge were over five times more likely to discuss PrEP (aOR: 5.10, 95% CI: 2.31–11.27) and more than three times more likely to prescribe it (aOR: 3.29, 95% CI: 1.72–6.30) compared with providers reporting lower knowledge levels (Table 3).

**Table 1. Demographic and professional characteristics of healthcare providers from North Louisiana (N = 102).**

| Variables | n | % |
|---|---|---|
| **Demographics** | | |
| Age, mean (SD) | 36.3 (±11.8) | — |
| Sex | | |
| Male | 44 | 43.1 |
| Female | 57 | 55.9 |
| Prefer not to answer | 1 | 1.0 |
| Race/Ethnicity | | |
| White | 66 | 64.7 |
| Black | 5 | 4.9 |
| Asian | 17 | 16.7 |
| Hispanic/Latino | 7 | 6.9 |
| Other | 7 | 6.9 |
| Position | | |
| Faculty | 29 | 28.4 |
| Resident | 63 | 61.8 |
| Fellow | 6 | 5.9 |
| Other | 4 | 3.9 |
| Specialty | | |
| Internal Medicine | 27 | 26.5 |
| Emergency Medicine | 18 | 17.6 |
| Pediatrics | 14 | 13.7 |
| Family Medicine | 15 | 14.7 |
| Obstetrics & Gynecology | 16 | 15.7 |
| Other | 12 | 11.8 |
| Years in practice | | |
| <5 years | 67 | 65.7 |
| 5–10 years | 13 | 12.7 |
| 11–15 years | 2 | 2.0 |
| 16–20 years | 3 | 2.9 |
| >20 years | 17 | 16.7 |

**\*** Abbreviations: PrEP, pre-exposure prophylaxis; SD, standard deviation.

### Provider knowledge and PrEP prescribing barriers

Greater knowledge was also associated with reduced odds of perceiving institutional barriers, including lack of training (aOR: 0.52, 95% CI: 0.32–0.85) and absence of protocols (aOR: 0.38, 95% CI: 0.23–0.63). In contrast, providers who believed that PrEP reduces patient sexual risk were more likely to report systemic limitations such as unclear clinical protocols and time or staffing constraints (Table 4).

## Discussion

### Key findings

This study identified substantial gaps in PrEP prescribing among healthcare providers in North Louisiana, driven primarily by limited provider knowledge and institutional barriers. These findings highlight the critical role of provider preparedness and

**Table 2. PrEP related practices and reported barriers among healthcare providers from North Louisiana (N = 102).**

| Variables | n | % |
|---|---|---|
| **Perceived patient HIV risk** | | |
| < 5% | 33 | 32.4 |
| 6–10% | 28 | 27.5 |
| 11–15% | 14 | 13.7 |
| >16% | 27 | 26.5 |
| **PrEP-related practices** | | |
| Patients asked about PrEP | | |
| No | 73 | 71.6 |
| Yes | 29 | 28.4 |
| Discussed PrEP with patient | | |
| No | 49 | 48.0 |
| Yes | 53 | 52.0 |
| Ever prescribed PrEP | | |
| No | 74 | 72.5 |
| Yes | 28 | 27.5 |
| Referred patient for PrEP | | |
| No | 70 | 68.6 |
| Yes | 32 | 31.4 |
| **Reported barriers to prescribing** | | |
| Lack of provider training | | |
| No | 16 | 15.7 |
| Yes | 86 | 84.3 |
| Lack of clinic protocols | | |
| No | 32 | 31.4 |
| Yes | 70 | 68.6 |
| Lack of monitoring resources | | |
| No | 37 | 36.3 |
| Yes | 65 | 63.7 |
| Staffing/time constraints | | |
| No | 31 | 30.4 |
| Yes | 71 | 69.6 |
| Insurance/cost barriers | | |
| No | 30 | 29.4 |
| Yes | 72 | 70.6 |
| **Provider perceptions of PrEP** | | |
| Perceived effectiveness | | |
| Slightly effective | 83 | 81.4 |
| Moderately effective | 18 | 17.6 |
| Very effective | 1 | 1.0 |
| Self-rated knowledge | | |
| Poor | 19 | 18.6 |
| Fair | 52 | 51.0 |
| Good | 19 | 18.6 |
| Very good | 4 | 3.9 |
| Excellent | 8 | 7.8 |

*(Continued)*

**Table 2.** (Continued)

| Variables | n | % |
|---|---|---|
| Belief PrEP promotes risky behavior | | |
| Disagree | 25 | 24.5 |
| Neither agree nor disagree | 22 | 21.6 |
| Agree | 55 | 53.9 |
| Belief PrEP reduces sexual risk | | |
| Not likely | 26 | 25.5 |
| Slightly likely | 35 | 34.3 |
| Likely | 31 | 30.4 |
| Very likely | 10 | 9.8 |

* Abbreviations: PrEP, pre-exposure prophylaxis; SD, standard deviation.

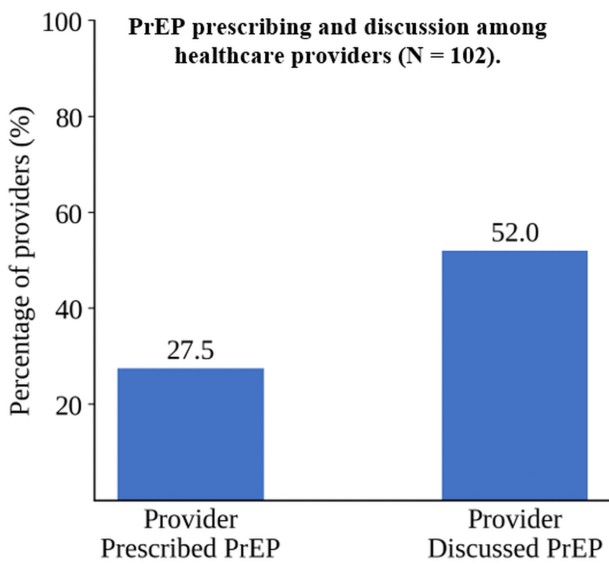

**Fig 1. PrEP prescribing and discussion among healthcare providers (N = 102).**

**Table 3. Multivariable linear and logistic regression predicting provider prescribing practices (N = 102).**

| Predictor | % patients at risk (adjusted B, SE) | Patients asked about PrEP (adjusted OR, 95% CI) | Discussed PrEP with patient (adjusted OR, 95% CI) | Ever prescribed PrEP (adjusted OR, 95% CI) | Referred patient for PrEP (adjusted OR, 95% CI) |
|---|---|---|---|---|---|
| PrEP effectiveness | −0.19 (0.30) | 2.39 (0.63-9.05) | 1.72 (0.55-5.33) | 0.45 (0.08-2.42) | 1.84 (0.62-5.45) |
| PrEP knowledge | 0.03 (0.12) | 4.96 (2.19-11.22)*** | 5.10 (2.31-11.27)*** | 3.29 (1.72-6.30)*** | 1.15 (0.72-1.85) |
| Belief PrEP promotes risky behavior | −0.21 (0.15) | 0.90 (0.46-1.78) | 1.16 (0.65-2.07) | 1.20 (0.60-2.36) | 0.67 (0.39-1.16) |
| Belief PrEP reduces sexual risk | 0.12 (0.13) | 2.93 (1.46-5.86)** | 1.13 (0.66-1.92) | 0.97 (0.54-1.72) | 0.92 (0.57-1.48) |

*Notes: B, unstandardized regression coefficient; SE, standard error; OR, odds ratio; CI, confidence interval. Models were adjusted for sex, age, race/ethnicity, hospital role, specialty, and years in practice. Statistical significance is indicated by asterisks and inferred from 95% confidence intervals (*$p < .05$; **$p < .01$; ***$p < .001$).

**Table 4.** Multivariable logistic regression predicting perceived barriers to prescribing PrEP (N = 102).

| Predictor | Lack of provider training (adjusted OR, 95% CI) | Lack of clinic protocols (adjusted OR, 95% CI) | Lack of monitoring resources (adjusted OR, 95% CI) | Staffing/time constraints (adjusted OR, 95% CI) | Insurance/cost barriers (adjusted OR, 95% CI) |
|---|---|---|---|---|---|
| PrEP effectiveness | 2.65 (0.32-22.28) | 3.63 (0.70-18.97) | 2.01 (0.64-6.34) | 0.52 (0.16-1.65) | 1.92 (0.56-7.15) |
| PrEP knowledge | 0.52 (0.32-0.85)** | 0.38 (0.23-0.63)*** | 1.08 (0.68-1.71) | 0.66 (0.39-1.12) | 0.78 (0.49-1.27) |
| Belief PrEP promotes risky behavior | 1.04 (0.52-2.08) | 1.14 (0.64-2.02) | 1.07 (0.63-1.83) | 0.76 (0.41-1.39) | 0.67 (0.37-1.21) |
| Belief PrEP reduces sexual risk | 1.22 (0.68-2.20) | 1.79 (1.04-3.08)* | 1.68 (1.05-2.69)* | 1.73 (1.03-2.93)* | 1.30 (0.80-2.11) |

*Notes: OR, odds ratio; CI, confidence interval. Models were adjusted for sex, age, race/ethnicity, hospital role, specialty, and years in practice. Statistical significance is indicated by asterisks and inferred from 95% confidence intervals (*$p < .05$; **$p < .01$; ***$p < .001$).

health system infrastructure in expanding access to HIV prevention. In a study among clinicians highly engaged in HIV care, only 54% had ever prescribed PrEP, indicating that even providers working with patients at risk for HIV face barriers to offering PrEP [8]. While providers in our study acknowledged that many of their patients may be at high risk for HIV, actual prescription rates of PrEP in Louisiana remain low [9]. Assessing HIV risk may often be dependent on patient-reported sexual risky behaviors, which may be limited by stigma or discomfort [9]. Moreover, relying on self-report and current clinical criteria can lead to systematic exclusion of vulnerable individuals who could benefit from PrEP [10]. Studies show that HIV risk alone does not lead to higher PrEP use unless supported by provider education and structural readiness for stigma-free PrEP conversations [2,3,11]. Our participants reported inadequate institutional support and the absence of standardized prescribing guidelines or systems, which is similar to results from previous studies in the South, as well as other research in England and Brazil [12].

Our findings reinforce the importance of improving healthcare providers' knowledge regarding PrEP. Providers with greater knowledge about PrEP were over five times more likely to discuss it with patients and three times more likely to prescribe it to them. These results align with previous studies, which have consistently shown that knowledge is a key facilitator of PrEP implementation [12–14]. A lower knowledge of prescribing guidelines, uncertainty of how to appropriately identify possible candidates for PrEP, and a lack of PrEP-specific training were also associated with decreased prescribing rates among healthcare providers due to inexperience with PrEP and other HIV medications. However, providers who showed increased familiarity with PrEP as a result of specialized training or patients requesting it, also showed decreased knowledge gaps concerning PrEP and increased rates of PrEP prescription [15,16].

## Policy and programmatic implications

Significantly, knowledge gaps were also tied to perceptions of institutional barriers, suggesting that institutional support may play a critical role in expanding PrEP uptake. Many healthcare providers in this study indicated insufficient training related to PrEP and identified the lack of clinical guidelines as a barrier. Previous research indicates the need for educational interventions, innovative approaches to the delivery of HIV care, and efforts to reduce stigma around PrEP [17]. Without these measures, PrEP utilization will remain low, especially among those who need it the most. Interdisciplinary care teams, PrEP navigators, and the development of standard prescribing protocols may also address these limitations [18,19]. Digital communications, like two-way digital messaging, have been recognized as best practices for enhancing provider-patient engagement and ensuring that patients continue taking PrEP [20,21]. Implementing a standard of quarterly clinical lab monitoring appointments in conjunction with virtual clinic visits provides a more patient-centered approach and may increase patient compliance with PrEP treatment [22]. Another strategy to overcome systemic barriers includes embedding prompts into electronic health records and integrating PrEP screening into routine sexual health checkups [23].

Occupational barriers preventing PrEP prescription have also been identified. High-demand, fast-paced clinical and hospital settings allow insufficient time for physicians to provide adequate education surrounding PrEP and HIV prevention, consult with insurance companies to establish coverage, and manage patient follow-up [24,25]. This is exacerbated in under-resourced areas of the Southern United States, where providers experience a large patient load and limited public health infrastructure [12,26]. One proposed solution to alleviate the time restriction is to further expand the roles of HIV and PrEP counselors and create a specific protocol concerning PrEP care to allocate roles to nurses, pharmacists, and other members of the healthcare team, which would reduce the time constraint placed on the physician [25].

Despite national initiatives such as the "Ready, Set, PrEP' program and Gilead's Advancing Access initiative, cost and insurance issues remain a barrier for providers to prescribe PrEP to their patients in Louisiana, as providers are often unaware of these programs [27,28]. The lack of comprehensive HIV prevention infrastructure has enabled a structural epidemic to continue, particularly among low-income, uninsured, and racial/ethnic minority populations [4]. These barriers reflect broader structural inequities that continue to shape HIV prevention efforts in the Southern United States. In states like Louisiana, underfunded public health systems and entrenched stigma have disproportionately impacted low-income communities and people of color. The National HIV/AIDS Strategy (NHAS) for 2022–2025 aims to eliminate the HIV epidemic in the United States by 2030. It requires a whole-of-government effort to address these disparities by integrating HIV prevention into initiatives that promote health equity and social justice [29]. These findings highlight the need for systemic interventions at the institutional level to improve PrEP prescribing via multitiered strategies that include provider training, streamlined protocols, and team-based models of care rather than solely relying on individual provider champions.

Moreover, interventions must address the intersectional realities of care delivery. For instance, obstetrics and gynecology providers may encounter cisgender women at risk for HIV, yet receive little to no PrEP-specific training [30]. Emergency department physicians also encounter at-risk patients, creating an opportunity for PrEP conversations and prescriptions; however, they lack the training and support to provide such services [31]. Future studies should investigate the interaction between provider identity and specialty on prescribing behaviors and attitudes.

Finally, the expansion of PrEP into non-traditional settings, including mobile clinics, pharmacies, telehealth platforms, and community-based organizations, has shown promise in addressing access inequities [23]. These models may be particularly effective in places like Louisiana, where a lack of healthcare and mistrust of the medical system further compound the risk of HIV transmission. Previous studies have shown that medical mistrust is associated with decreased PrEP awareness, willingness to try PrEP, and adherence, with targeted interventions showing an increase in positive outcomes among minority populations [32]. Other identified barriers to PrEP access include decreased access to PrEP providers, inadequate health insurance or financial resources, and limited health literacy [5]. Community programs and mobile PrEP clinics that offer affordable, low-cost, or no-cost services have been successful in alleviating some of the identified barriers to patients' knowledge of PrEP and access to PrEP prescriptions and treatment [33].

## Limitations and future research

Several limitations should be considered when interpreting these findings. First, due to the cross-sectional design, causal inferences cannot be established. Second, self-reported data on prescribing practices and attitudes are subject to social desirability and recall bias. However, anonymous survey administration may have reduced the likelihood of socially desirable responses. Additionally, findings may not be generalizable beyond academic-affiliated providers in the Southern United States. Finally, while the survey assessed key provider- and system-level barriers, it did not include qualitative data that could further elucidate institutional culture, stigma, or provider–patient communication dynamics. Future research should incorporate multi-site designs, qualitative methods, and longitudinal approaches to understand how organizational interventions influence prescribing PrEP across the Southern United States.

## Conclusion

This study highlights critical gaps in PrEP knowledge, prescribing practices, and structural readiness among healthcare providers in North Louisiana. To close these gaps and reduce HIV transmission in high-burden regions, targeted policies are essential. We recommend mandatory PrEP training as part of continuing education, the deployment of trained PrEP navigators, and expanded access through telehealth and pharmacy-based PrEP services. Additionally, embedding HIV risk prompts and PrEP order sets in electronic health records, removing insurance and cost-related barriers, and publicly reporting PrEP coverage by race and region can improve equitable access. Addressing stigma, enhancing provider confidence, and tailoring strategies to vulnerable populations are all essential steps in ensuring that PrEP fulfills its potential as a powerful tool in ending the HIV epidemic by 2030 [34].

## Supporting information

**S1 Dataset. De-identified dataset underlying the findings of this study.** The file contains survey responses from 102 healthcare providers in North Louisiana used for the analyses reported in the manuscript.
(CSV)

## Acknowledgments

We would like to acknowledge the assistance of Mr. Elliott Freeman, Head Writing and Publishing Librarian at the LSU Health Sciences Center-Shreveport, for reviewing and editing this article.

## Author contributions

**Conceptualization:** Deborah G. Smith.

**Data curation:** Deborah G. Smith, Camila Castro.

**Formal analysis:** Luis Enrique Espinoza.

**Investigation:** Deborah G. Smith, Alexandre Malek.

**Methodology:** Deborah G. Smith, Patricia Pichilingue-Reto.

**Project administration:** Deborah G. Smith.

**Supervision:** Deborah G. Smith.

**Writing – original draft:** Deborah G. Smith, Camila Castro, Patricia Pichilingue-Reto, Catherine G. Dean, Alexandre Malek, Luis Enrique Espinoza.

**Writing – review & editing:** Deborah G. Smith, Camila Castro, Patricia Pichilingue-Reto, Catherine G. Dean, Luis Enrique Espinoza.

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
