## [Decision Letter · Decision Letter 0]

18 Jan 2026

PONE-D-25-49893Knowledge Gaps and Structural Barriers to Prescribing Pre-Exposure Prophylaxis Among Healthcare Providers in North Louisiana: A Cross-Sectional StudyPLOS One

Dear Dr. Smith,

Thank you for submitting your manuscript to PLOS ONE. After careful consideration, we feel that it has merit but does not fully meet PLOS ONE’s publication criteria as it currently stands. Therefore, we invite you to submit a revised version of the manuscript that addresses the points raised during the review process.

We recommend that you respond to all of the reviewers' comments and make the necessary changes to the manuscript to improve its quality. My specific feedback from my evaluation of the manuscript is on the method section of which I recommend you add more relevant details that can help others who might wish to replicate the study. Secondly, on the lines 114-115, you mentioned the use of Linear regression, and you used it to regress other variables on "Perceived patient HIV risk". The outcome variable seems more like a categorical variable, than a continuous quantitative variable. However, if you consider it a continuous quantitative variable, did it fulfil the assumptions (e.g. linearity) guiding Linear regression before doing the regression analysis? Otherwise, I recommend you recategorize it into a binary variable and use logistic regression for the modelling.  Please submit your revised manuscript by Mar 04 2026 11:59PM. If you will need more time than this to complete your revisions, please reply to this message or contact the journal office at plosone@plos.org. Please include the following items when submitting your revised manuscript:

We look forward to receiving your revised manuscript.

Kind regards,

Ayodeji Babatunde Oginni

Academic Editor

PLOS One

Journal Requirements:

2. Please amend your authorship list in your manuscript file to include author Luis E Espinoza.

3. Please amend the manuscript submission data (via Edit Submission) to include author Luis Enrique Espinoza.

4. We are unable to open your Supporting Information file [HIV data Mar 21.sav.zip]. Please kindly revise as necessary and re-upload.

Reviewers' comments:

Reviewer's Responses to Questions

**Comments to the Author**

1. Is the manuscript technically sound, and do the data support the conclusions?

Reviewer #1: Yes

Reviewer #2: Partly

2. Has the statistical analysis been performed appropriately and rigorously? 

Reviewer #1: Yes

Reviewer #2: Yes

3. Have the authors made all data underlying the findings in their manuscript fully available?

Reviewer #1: Yes

Reviewer #2: Yes

4. Is the manuscript presented in an intelligible fashion and written in standard English?

Reviewer #1: Yes

Reviewer #2: Yes

5. Review Comments to the Author

Reviewer #1: 1. Title and Abstract

• The abstract includes excessive background information but lacks a concise statement of study objectives and key results. Include a clear objective statement in the abstract and emphasize key findings with numerical data (e.g., “27.5% had prescribed PrEP; 84% cited lack of training”).

• Statistical results are presented descriptively without emphasizing the main significant findings. Conclude with a brief, action-oriented implication: “Targeted provider training and institutional policy support are essential to improving PrEP implementation in high-burden settings.”

2. Introduction

• The background is comprehensive but somewhat repetitive. Transitions between global, national, and regional contexts are abrupt. Consolidate background information to three key paragraphs: 1) Burden and relevance of PrEP, 2) Known provider-level barriers, and 3) Specific knowledge gap in Louisiana.

• The research gap and study objectives are not clearly articulated. End the section with a concise objective statement: “This study aims to assess knowledge, attitudes, and structural barriers influencing PrEP prescribing among healthcare providers in North Louisiana.”

3. Materials and Methods

• The subheading “Study setting and settings” should be corrected to “Study Design and Setting.”

• The survey instrument’s validation process (pilot testing, reliability) is not described. Describe the survey instrument development, including question sources, piloting, and validation (if applicable).

• Data management and handling of missing values are unclear. Include details on data handling: “Incomplete responses were excluded using listwise deletion, resulting in a final analytic sample of N=___.”

4. Results

• Tables are overcrowded and results are repeated in both text and tables. Present only key findings in the text, referring readers to tables for details. Reorganize Table 1 into two tabless: (1) demographic characteristics and (2) PrEP-related behaviors/barriers.

• Figures are limited; visual aids would help communicate key findings. Include one or two visualizations (e.g., a bar chart showing the proportion of providers who prescribed or discussed PrEP).

5. Discussion

• Restructure discussion into clear subsections: 1) Key findings summary, 2) Interpretation in context of existing literature, 3) Policy and programmatic implication, and 4) Study limitations and future research.

• Remove redundant or overly descriptive paragraphs.

• Strengthen interpretation by connecting findings to potential interventions: “Limited provider training and absence of clinic-level protocols suggest that institutional support mechanisms, such as PrEP champions or teleconsultation models, could facilitate broader PrEP uptake.”

• Include a dedicated limitations paragraph addressing cross-sectional design, self-report bias, and lack of qualitative exploration.

6. Conclusion

• Make the conclusion shorter and more impactful: “This study highlights persistent provider knowledge gaps and structural barriers to PrEP prescribing in North Louisiana. Enhancing provider education, developing institutional protocols, and addressing systemic barriers such as stigma and cost are essential steps toward equitable PrEP implementation.”

Reviewer #2: Hello,

The title sounds good, but my biggest concern on the discussion part you have discussed based on local context not comparing on the studies around the world. Also on the methodology part especially sampling were not clear as well as sample size no clear how was it obtained.

6. PLOS authors have the option to publish the peer review history of their article (what does this mean?). If published, this will include your full peer review and any attached files.

Reviewer #1: **Yes:** Faizul Akmal Abdul Rahim

Reviewer #2: No

---

## [Author Response · Author response to Decision Letter 1]

20 Feb 2026

Response to the academic editor and reviewers

Dear Academic Editor and Reviewers,

We sincerely thank you for your thoughtful evaluation of our manuscript and for your constructive feedback. We have carefully revised the manuscript to address all comments and believe these revisions have strengthened the clarity, methodological transparency, and overall quality of the paper. Below, we provide a detailed, point-by-point response to each comment.

Academic Editor Comments

Comment 1: My specific feedback from my evaluation of the manuscript is on the method section of which I recommend you add more relevant details that can help others who might wish to replicate the study.

Response: Thank you for this important suggestion. We expanded the Methods section to provide additional detail regarding recruitment procedures, including the use of QR codes and electronic survey links, participant information and consent procedures, eligibility screening, and automated survey flow.

Comment 2: Secondly, on the lines 114-115, you mentioned the use of Linear regression, and you used it to regress other variables on "Perceived patient HIV risk". The outcome variable seems more like a categorical variable, than a continuous quantitative variable. However, if you consider it a continuous quantitative variable, did it fulfil the assumptions (e.g. linearity) guiding Linear regression before doing the regression analysis? Otherwise, I recommend you recategorize it into a binary variable and use logistic regression for the modelling.

Response: The variable representing the percentage of patients perceived to be at risk for HIV was originally measured using ordered response categories. Ordinal logistic regression was initially considered; however, testing indicated that the proportional odds assumption was violated. Collapsing the variable into a binary outcome would have resulted in loss of information and reduced variability. Therefore, the variable was modeled as continuous using linear regression, an accepted approach for ordinal variables with multiple ordered categories when assumptions are reasonably met. Model diagnostics were examined to confirm that linear regression assumptions were adequately satisfied. We have clarified this analytic decision in the Methods section.

Authorship Amendments

Comment: Please amend authorship to include Luis Enrique Espinoza.

Response: The authorship list and submission metadata have been updated to reflect the author’s full name, Luis Enrique Espinoza.

Supporting Information File

Comment: Unable to open supporting file.

Response: We have replaced the file and uploaded a revised version.

Supporting Information Captions

Response: Captions for Supporting Information file and Figure have been added at the end of the manuscript, and in-text citations have been updated accordingly.

Reference List Review

Response: We reviewed the reference list for completeness and accuracy, removed duplicate citations, and added relevant references where appropriate, based on the reviewers' recommendations.

Response to Reviewer #1

Title and Abstract

Comment: The abstract includes excessive background information but lacks a concise statement of study objectives and key results. Include a clear objective statement in the abstract and emphasize key findings with numerical data (e.g., “27.5% had prescribed PrEP; 84% cited lack of training”).

Response: We revised the abstract to substantially reduce background content, added a clear objective statement, and emphasized key findings with numerical data.

Comment: Statistical results are presented descriptively without emphasizing the main significant findings. Conclude with a brief, action-oriented implication: “Targeted provider training and institutional policy support are essential to improving PrEP implementation in high-burden settings.”

Response: We revised the abstract conclusion to highlight the primary findings and added a concise, action-oriented implication that emphasizes the importance of targeted provider training and supportive institutional policies to improve PrEP implementation in high-HIV-burden Southern settings.

Introduction

Comment: The background is comprehensive but somewhat repetitive. Transitions between global, national, and regional contexts are abrupt. Consolidate background information to three key paragraphs: 1) Burden and relevance of PrEP, 2) Known provider-level barriers, and 3) Specific knowledge gap in Louisiana.

Response: We reorganized the Introduction into three focused paragraphs addressing (1) the burden and relevance of PrEP, (2) known provider-level barriers, and (3) the research gap and study objective. Transitions were strengthened, and a clear objective statement was added at the end of the section.

Comment: The research gap and study objectives are not clearly articulated. End the section with a concise objective statement: “This study aims to assess knowledge, attitudes, and structural barriers influencing PrEP prescribing among healthcare providers in North Louisiana.”

Response: We added the objective recommended by Reviewer 1 to the end of the introduction.

Materials and Methods

Comment: The subheading “Study setting and settings” should be corrected to “Study Design and Setting.”

Response: The subheading was revised to “Study Design and Setting.”

Comment: The survey instrument’s validation process (pilot testing, reliability) is not described. Describe the survey instrument development, including question sources, piloting, and validation (if applicable).

Response: We expanded the Methods section to describe the survey development process. Items were adapted from previously published studies, and the instrument was pilot-tested among physicians to assess clarity, readability, and flow. Minor revisions were made prior to distribution, and pilot participants were excluded from the analytic sample.

Comment: Data management and handling of missing values are unclear. Include details on data handling: “Incomplete responses were excluded using listwise deletion, resulting in a final analytic sample of N=___.”

Response: We clarified that incomplete responses were excluded using listwise deletion, resulting in a final analytic sample of 102 providers.

Results

Comment: Tables are overcrowded and results are repeated in both text and tables. Present only key findings in the text, referring readers to tables for details. Reorganize Table 1 into two tabless: (1) demographic characteristics and (2) PrEP-related behaviors/barriers.

Response: We revised the Results section to present only key findings in the text while directing readers to tables for detailed data. Table 1 was divided into two tables to improve readability.

Comment: Figures are limited; visual aids would help communicate key findings. Include one or two visualizations (e.g., a bar chart showing the proportion of providers who prescribed or discussed PrEP).

Response: We added Figure 1 to visually highlight PrEP prescribing, discussion practices, and training barriers.

Discussion

Comment: Restructure discussion into clear subsections: 1) Key findings summary, 2) Interpretation in context of existing literature, 3) Policy and programmatic implication, and 4) Study limitations and future research.

Response: The Discussion was reorganized into clearly labeled subsections addressing key findings, interpretation within the existing literature, policy and programmatic implications, and study limitations, along with directions for future research.

Comment: Remove redundant or overly descriptive paragraphs.

Response: We removed overly descriptive paragraphs.

Comment: Strengthen interpretation by connecting findings to potential interventions: “Limited provider training and absence of clinic-level protocols suggest that institutional support mechanisms, such as PrEP champions or teleconsultation models, could facilitate broader PrEP uptake.”

Response: We strengthened the discussion by explicitly linking the findings to potential institutional and provider-level interventions and added a dedicated limitations paragraph that addressed the cross-sectional design, self-report bias, and generalizability.

Comment: Include a dedicated limitations paragraph addressing cross-sectional design, self-report bias, and lack of qualitative exploration.

Response: We added a dedicated limitations paragraph that addressed the cross-sectional design, self-report bias, and generalizability

Conclusion

Comment: Make the conclusion shorter and more impactful: “This study highlights persistent provider knowledge gaps and structural barriers to PrEP prescribing in North Louisiana. Enhancing provider education, developing institutional protocols, and addressing systemic barriers such as stigma and cost are essential steps toward equitable PrEP implementation.”

Response: The conclusion was revised for clarity and concision, incorporating the reviewer’s suggestion.

Response to Reviewer #2

Comment: The title sounds good, but my biggest concern on the discussion part you have discussed based on local context not comparing on the studies around the world. Also on the methodology part especially sampling were not clear as well as sample size no clear how was it obtained.

Response: We revised the Discussion to incorporate comparisons with studies conducted in other regions and countries, situating our findings within the broader global literature while maintaining regional relevance.

We also clarified the sampling strategy and recruitment procedures. Eligible physician faculty members, residents, and fellows were recruited using convenience sampling with the goal of reaching all providers affiliated with the medical school and teaching hospital during the study period. Additionally, we specified that 104 providers completed the survey and that incomplete responses were excluded using listwise deletion, resulting in a final analytic sample of 102 providers.

Response to Reviewer #3

Abstract

Comment: Write the section of abstract into structured paragraphs.

Response: We revised the abstract by organizing it into clearly labeled sections (Background, Objective, Methods, Results, and Conclusions).

Comment: Include p-values of the variables which showed significance

Response: We present adjusted effect estimates with 95% confidence intervals, which provide information on both the direction and precision of the associations while maintaining clarity in the abstract.

Introduction

Comment: State in number rates of new HIV diagnoses in the country

Response: We stated the rates of new HIV diagnoses in the country.

Comment: State situation globally, regional and locally as far as Pre- exposure prophylaxis is concern

Response: We strengthened the Introduction by incorporating global, national, regional, and state-level epidemiologic data and by adding context on disparities in PrEP uptake.

Methodology

Comment: State the high HIV burden in the region(North Louisiana).

Response: We added epidemiologic context to describe Louisiana as a high-HIV-burden state to better justify the study setting.

Comment: What is the ideal sampling technique you have used in the selection of study participants.

Response: We clarified that participants were recruited using convenience sampling.

Comment: Why did you remained with only 102 out 104?

Response: We clarified that incomplete responses were excluded using listwise deletion, resulting in a final sample of 102 providers.

Measures

Comment: Where did you adapt this set of questions? How did you measure knowledge and practice without using scale?

Response: We clarified that the questionnaire was informed by previously published studies examining providers’ knowledge, attitudes, and prescribing behaviors related to PrEP and HIV prevention. Provider knowledge was assessed through self-reported familiarity with PrEP, awareness of prescribing guidelines, and prior training, while prescribing practices were evaluated using items assessing PrEP discussion, prescribing history, and confidence in identifying eligible patients.

Comment: What were the inclusion criteria? What were the exclusion criteria?

Response: We added both inclusion and exclusion criteria.

Comment: Show what was the collection tools used in this study?? It is not clear stating electronically

Response: We added to the data management and analysis subsection that “Survey data were collected electronically using REDCap (Research Electronic Data Capture), a secure web-based platform designed to support research data collection.”

Ethical Approval

Comment: Show how the confidentiality and privacy was done as well as how participants were enrolled ethically.

Response: We revised the Ethics section to provide additional detail on IRB approval, informed consent waivers, voluntary participation, and data confidentiality procedures. The survey was anonymous, no personally identifiable information was collected, and all data were stored on secure, password-protected servers accessible only to the research team.

Results

Comment: Include the p-values for the significantly associated variables

Response: We included under the tables 3 and 4 that “Statistical significance is shown by asterisks and derived from 95% confidence intervals (*p < .05; **p < .01; ***p < .001).”

Comment: Provide N within each percentage composition in the results

Response: We included the n within each percentage in the results.

Discussion

Comment: Please compare findings with other studies around the world

Response: We expanded the Discussion to include comparisons with international studies.

Comment: Please provide countries and year of study in which specific studies were conducted

Response: We provided the countries in the discussion, and the study years are reflected in the cited literature.

Comment: Provide long form of OB/GYN

Response: We provided the long form of OB/GYN: obstetrics and gynecology (see line 466)

Conclusion

Comment: Provide recommendations separately.

Response: We added a section outlining recommendations for practice.

General Comments

Comment: Do you think 102 participants are essentially enough for publication and later be generalizable to the country.

Response: The primary goal of this study was to provide a detailed, regionally focused assessment within a high HIV-burden Southern setting rather than achieve national generalizability. We clarified in the Discussion that findings should be interpreted within this regional context.

Comment: Why was study not a mixed design???

Response: This study was designed as a cross-sectional quantitative survey to systematically assess provider-level knowledge gaps, prescribing practices, and structural barriers. We agree that qualitative methods could provide additional contextual insight and noted in the Discussion that future research should incorporate mixed-methods approaches.

Comment: What were the limitations of the study and how to mitigate them

Response: We added a dedicated limitations subsection that describes the cross-sectional design, reliance on self-reported data, and the lack of qualitative exploration. Anonymous data collection was used to reduce reporting bias, and future longitudinal and mixed-methods research is recommended.

Comment: Provide each authors contribution as well

Response: Author contributions have been added.

---

## [Editor Report · Decision Letter 1]

22 Mar 2026

PONE-D-25-49893R1Knowledge Gaps and Structural Barriers to Prescribing Pre-Exposure Prophylaxis Among Healthcare Providers in North Louisiana: A Cross-Sectional StudyPLOS One

Dear Dr. Smith,

Thank you for submitting your manuscript to PLOS ONE. After careful consideration, we feel that it has merit but does not fully meet PLOS ONE’s publication criteria as it currently stands. Therefore, we invite you to submit a revised version of the manuscript that addresses the points raised during the review process. **I would like you to include under the "Participant and Recruitment" section, the sample size calculated or estimated for the study, including your assumptions.**

We look forward to receiving your revised manuscript.

Kind regards,

Ayodeji Babatunde Oginni

Academic Editor

PLOS One
---

## [Author Response · Author response to Decision Letter 2]

23 Mar 2026

Dear reviewers and editor,

Thank you for your question. We agree that reporting how the sample size was estimated would improve the transparency and rigor of our methodology. We have added the following paragraph and a new reference used to estimate the study sample size required for our study, and we have included details in the “Participants and Recruitment” section.

“Although convenience sampling was employed within a single academic medical center, an estimated sample size was considered to ensure adequate precision of proportion estimates. Since there were no prior estimates of the proportion of providers reporting the outcomes of interest (e.g., PrEP prescribing), a conservative assumed proportion of 50% was used, as recommended by Lwanga and Lemeshow [7], in standard sample size formulas for estimating proportions in cross-sectional studies, because this produces the largest required sample size. Assuming a 95% confidence level and a margin of error of ±10 percentage points, the minimum estimated sample size required was approximately 96 participants. A total of 104 healthcare providers completed the survey. Incomplete responses were excluded using listwise deletion, resulting in a final analytic sample of 102.”

Best regards,

Dr. Smith

---

## [Editor Report · Decision Letter 2]

26 Mar 2026

Knowledge Gaps and Structural Barriers to Prescribing Pre-Exposure Prophylaxis Among Healthcare Providers in North Louisiana: A Cross-Sectional Study

PONE-D-25-49893R2

Dear Dr. Smith,

We’re pleased to inform you that your manuscript has been judged scientifically suitable for publication and will be formally accepted for publication once it meets all outstanding technical requirements.

Kind regards,

Ayodeji Babatunde Oginni

Academic Editor

PLOS One
---

## [Editor Report · Acceptance letter]

PONE-D-25-49893R2

PLOS One

Dear Dr. Smith,

I'm pleased to inform you that your manuscript has been deemed suitable for publication in PLOS One. Congratulations! Your manuscript is now being handed over to our production team.

Kind regards,

on behalf of

Ayodeji Babatunde Oginni

Academic Editor

PLOS One